Taxonomic overview of the hyperdiverse ant genus Tetramorium Mayr (Hymenoptera, Formicidae) in India with descriptions and X-ray microtomography of two new species from the Andaman Islands

http://orcid.org/0000-0003-3977-3775 Agavekar Gaurav 1 2 3 gauravga@ncbs.res.in
http://orcid.org/0000-0003-4709-3083 Hita Garcia Francisco 3
Economo Evan P. 3
1 Master’s Program in Wildlife Biology and Conservation, Wildlife Conservation Society – India Program and National Centre for Biological Sciences , Bangalore , India
2 National Centre for Biological Sciences, Tata Institute of Fundamental Research , Bangalore , India
3 Okinawa Institute of Science and Technology Graduate University , Onna-son, Okinawa , Japan
Ward Darren
Electronic publication date: 2017 Sep 20
Publication date: 2017
Volume: 5
Electronic Location ID: e3800
Received 2017 Jun 30; Accepted 2017 Aug 22
Copyright: © 2017 Agavekar et al.
Copyright year: 2017
Copyright holder: Agavekar et al.
License: This is an open access article distributed under the terms of the Creative Commons Attribution License, which permits unrestricted use, distribution, reproduction and adaptation in any medium and for any purpose provided that it is properly attributed. For attribution, the original author(s), title, publication source (PeerJ) and either DOI or URL of the article must be cited.
License URL: https://creativecommons.org/licenses/by/4.0/

Keywords: Micro-CT, Cybertype, Tetramorium inglebyi group, Tetramorium tonganum group, Ants, Biodiversity, Islands

Funding: Department of Science and Technology, Gov. of India National Centre for Biological Sciences Okinawa Institute of Science and Technology This work was funded by the Department of Science and Technology, Gov. of India; National Centre for Biological Sciences; and Okinawa Institute of Science and Technology. GA, FHG, and EPE were supported by subsidy funding to OIST. GA was also supported by a WCS-NCBS Master’s fellowship and a DST Inspire Faculty Award (IFA-13 LSBM-64) to D. Agashe during this research. The funders had no role in study design, data collection and analysis, decision to publish, or preparation of the manuscript.

==============================
With 600 described species, the ant genus Tetramorium represents one of the most species-rich ant radiations. However, much work remains to fully document the hyperdiversity of this remarkable group. Tetramorium, while globally distributed, is thought to have originated in the Afrotropics and is particularly diverse in the Old World. Here, we focus attention on the Tetramorium fauna of India, a region of high biodiversity value and interest for conservation. We overview Tetramorium diversity in India by providing a species list, accounts of all species groups present, an illustrated identification key to Indian Tetramorium species groups and notes on the Indian Tetramorium fauna. Further, we describe two new species, Tetramorium krishnani sp. n. and Tetramorium jarawa sp. n. from the Andaman Islands archipelago and embed them into currently recognized Tetramorium tonganum and Tetramorium inglebyi species groups. We also provide illustrated species level keys for these groups. Along with detailed species descriptions and high-resolution montage images of types, we provide 3D cybertypes of the new species derived from X-ray micro-computed tomography.

Introduction

The hyperdiverse, globally distributed ant genus Tetramorium Mayr is one of the largest ant genera with almost 600 extant and two fossil species described so far (Bolton, 2016). Most of its diversity is distributed in the Old World tropics and subtropics while very few species, mostly introduced non-natives, occur in the New World (Bolton, 1976, 1977, 1979, 1980; Brown, 1958). The Afrotropical region is believed to be the origin and center of diversity of the genus, from where ca. 230 valid and potentially 150 undescribed species are known (Bolton, 1976; Hita Garcia, Fischer & Peters, 2010; Hita Garcia & Fisher, 2014a). The Indomalayan region also harbors a rich Tetramorium fauna with approximately 120 species (Janicki et al., 2016; Guénard et al., 2017), of which 42 have been reported from India (Bharti et al., 2016; Bharti & Kumar, 2012).

The taxonomic foundation for the genus is in a relatively good condition thanks to the initial comprehensive taxonomic treatments by Bolton (1976, 1977, 1979, 1980) who revised all the regions except for the Palearctic. These works were the foundation for recent revisionary works focusing on the Afrotropical (Hita Garcia & Fischer, 2014; Hita Garcia, Fischer & Peters, 2010; Hita Garcia & Fisher, 2013, 2014a), Malagasy (Hita Garcia & Fisher, 2011, 2012a, 2012b, 2014b, 2015), and Palearctic regions (Csösz, Radchenko & Schulz, 2007; Csösz & Schulz, 2010). Nevertheless, the Indomalayan Tetramorium fauna as a whole or in parts was not revised since Bolton (1977), and our knowledge of Indomalayan Tetramorium diversity has only slightly grown through some smaller regional treatments or insular single species descriptions (Bharti, 2011; Bharti & Kumar, 2012; Schlick-Steiner, Steiner & Zettel, 2006; Sheela & Narendran, 1998; Sorger, 2011; Yamane & Jaitrong, 2011). Despite this, compared to most of the Indomalayan region, the Tetramorium fauna of India is much better studied. On the basis of Bolton (1977), eight new species have been described in the last two decades (Bharti, 2011; Bharti & Kumar, 2012; Mathew & Tiwari, 2000; Sheela & Narendran, 1998), which increased the species count to 40. Yet, the Indian subcontinent is vast and comprises an extraordinary diversity of landscapes, climate zones, and ecosystems, but only a small fraction of habitats has been sampled well and our knowledge of ant diversity and distribution is fragmentary and will increase with future collections.

We conducted an island-wide survey of ants on Havelock Island, part of the Andaman Islands archipelago. This tropical archipelago has a humid, warm climate and experiences heavy rainfall from southwest and northeast monsoons. Geographically, it is located in the Bay of Bengal, with mainland India to the west and Myanmar to the north and east. While the archipelago is administered by the Government of India, it is geographically much closer to Southeast Asia than mainland India. As a result, flora and fauna there show affinities to both SE Asian and mainland Indian elements. The islands harbor an impressive diversity of life forms with over a quarter of the archipelago’s flora and fauna believed to be endemic (Rao, Chandra & Devi, 2013). Detailed faunal surveys of the islands are generally lacking but some groups such as birds and butterflies are relatively well documented, albeit in a biogeographical context (Davidar et al., 2002). The ant fauna of these islands is poorly known, with the most recent field and literature survey reporting a total of 125 species (Mohanraj, Ali & Veenakumari, 2010). This number is undoubtedly an underestimate given the climatic and geographic setting of the islands and also due to the lack of diversity in sampling methods used in previous surveys. During our surveys, we mainly focused on leaf-litter ant communities and exhaustively sampled Havelock Island using Winkler leaf-litter extraction transects, as well as hand collection. Winkler extraction is the most efficient technique for the study of leaf-litter ants since it captures a greater proportion of ant species compared to other sampling methods (Fisher, 1999; Olson, 1991) and thus our surveys are among the first detailed surveys of ant fauna on these islands.

In this paper, we provide a taxonomic overview of the genus Tetramorium in India. We describe two new species of Tetramorium from Havelock Island and embed them into the existing species group system. In order to improve the taxonomy of the Indian Tetramorium fauna and facilitate classification of any future findings, we also provide accounts of all species groups present in India, an illustrated identification key to Indian Tetramorium species groups, and an updated species list for the region. Along with detailed species descriptions, we provide high-resolution montage images and illustrated identification keys.

Adding a third dimension to species documentation, we provide cybertypes of the new species by leveraging X-ray microtomography (micro-CT) technology to construct 3D surface models of the holotypes of the new species. Micro-CT is a non-invasive imaging technology that allows generation of high-resolution 3D reconstructions consisting of powerful and accurate representation of morphological and anatomical features of organisms being studied (Faulwetter et al., 2013; Friedrich et al., 2014). By enabling users to rotate, measure, section, and dissect virtually any part of the organism under study, such 3D models open up possibilities of detailed morphological and anatomical analyses, which would otherwise be difficult or impossible to perform. This technology has proved highly useful in a variety of research areas in biology, including comparative and functional morphology (Beutel, Ge & Hörnschemeyer, 2008; Metscher, 2009a; Wirkner & Prendini, 2007; Zimmermann et al., 2011), paleontological and forensic entomology (Barden & Grimaldi, 2012; Dierick et al., 2007; Richards et al., 2012), and developmental biology (Metscher, 2009b). Taxonomists have been relatively late in leveraging micro-CT, although lately it has gained momentum in invertebrate taxonomy for taxa as diverse as spiders (Michalik & Ramírez, 2013), earthworms (Fernández et al., 2014), flatworms (Carbayo, Francoy & Giribet, 2016; Carbayo & Lenihan, 2016), and myriapods (Akkari, Enghoff & Metscher, 2015; Stoev et al., 2013). Only a handful of studies have utilized it in insect taxonomy, specifically in butterflies and moths (Simonsen & Kitching, 2014) and ants (Fischer, Sarnat & Economo, 2016; Hita Garcia et al., 2017a, 2017b; Sarnat, Fischer & Economo, 2016). Our work thus adds to the growing number of studies employing micro-CT in invertebrate taxonomy and represents one of the few studies having applied it to insect taxonomy so far. While the usefulness of this technology in ant taxonomy has been discussed elsewhere in detail (Hita Garcia et al., 2017a, 2017b), cybertypes of new species allow examination of morphological characters in great detail and virtually eliminate the need to exchange holotypes among taxonomists. To this end, we provide micro-CT based still images as well as 3D rotation videos and 3D PDFs of both holotypes. The complete datasets containing the raw micro-CT data, 3D PDFs, 3D rotation videos, still images of 3D models, and color montage photos are made available online (Figshare, https://figshare.com) as cybertypes. In addition to the cybertype data at Figshare, we also provide freely accessible 3D surface models of both holotypes on Sketchfab (https://sketchfab.com/arilab).

Material Examined and Terminology

The material upon which this study is based is located at Research Collection Facility of the National Center for Biological Sciences, Bangalore, India (repository code: NCBS (Evenhuis, 2013)) and Zoological Survey of India (ZSI), Kozhikode, India. The new material examined in this study was collected during island-wide surveys of Havelock Island in late 2015 and early 2016. Field research and collection permits for the surveys were provided by the Department of Environment and Forest, Andaman and Nicobar Administration, Government of India (Permit no. CWLW/WL/134/353).

Havelock is a relatively small island (110 km2) with two main forest types distributed adjacent to each other. Evergreen forests are distributed inland and have dense canopy whereas the littoral forests are distributed along the coast and are characterized by relatively sparse canopy, high wind, and sandy soil. We focused on leaf-litter ant communities by sampling 35 transects each of length 80 m that were laid in the two forest types. On each transect, leaf-litter was collected from 5 × 1 m2 quadrats placed equidistantly from each other.

The general terminology for ant morphology predominantly follows Bolton (1980), Keller (2011), and Hita Garcia & Fischer (2014). The terminology for the description of surface sculpturing follows Harris (1979), and the description of degrees of inclination of pilosity follows Wilson (1955).

Measurements and indices

Morphometric measurements were performed with a Leica M165 microscope equipped with an orthogonal pair of micrometers at magnifications ranging from 60 to 100×. These measurements and indices are presented as minimum and maximum values with holotype measurements in parentheses. All measurements were recorded in mm to three decimal places, but are expressed in the study to two decimal places. The measurements and indices given below (Fig. 1) follow Hita Garcia & Fischer (2014): HL Head length: maximum distance from the midpoint of the anterior clypeal margin to the midpoint of the posterior margin of head, measured in full-face view. Impressions on the anterior clypeal margin and the posterior head margin reduce head length.

HW Head width: width of the head directly behind the eyes measured in full-face view.

SL Scape length: maximum scape length excluding basal condyle and neck.

EL Eye length: maximum diameter of compound eye measured in oblique lateral view.

PH Pronotal height: maximum height of the pronotum measured in lateral view.

PW Pronotal width: maximum width of the pronotum measured in dorsal view.

WL Weber’s length: diagonal length of the mesosoma in lateral view from the posteroventral margin of propodeal lobe to the anteriormost point of pronotal slope, excluding the neck.

PSL Propodeal spine length: in dorsofrontal view the tip of the measured spine, its base, and the center of the propodeal concavity between the spines must all be in focus. Using a dual-axis micrometer the spine length is measured from the tip of the spine to a virtual point at its base where the spine axis meets orthogonally with a line leading to the median point of the concavity.

PTH Petiolar node height: maximum height of the petiolar node measured in lateral view from the highest (median) point of the node to the ventral outline. The measuring line is placed at an orthogonal angle to the ventral outline of the node.

PTL Petiolar node length: maximum length of the dorsal face of the petiolar node from the anterodorsal to the posterodorsal angle, measured in dorsal view excluding the peduncle.

PTW Petiolar node width: maximum width of the dorsal face of the petiolar node measured in dorsal view.

PPH Postpetiole height: maximum height of the postpetiole measured in lateral view from the highest (median) point of the node to the ventral outline. The measuring line is placed at an orthogonal angle to the ventral outline of the node.

PPL Postpetiole length: maximum length of the postpetiole measured in dorsal view.

PPW Postpetiole width: maximum width of the postpetiole measured in dorsal view.

OI Ocular index: EL/HW * 100

CI Cephalic index: HW/HL * 100

SI Scape index: SL/HW * 100

DMI Dorsal mesosoma index: PW/WL * 100

LMI Lateral mesosoma index: PH/WL * 100

PSLI Propodeal spine index: PSL/HL * 100

PeNI Petiolar node index: PTW/PW * 100

LPeI Lateral petiole index: PTL/PTH * 100

DPeI Dorsal petiole index: PTW/PTL * 100

PpNI Postpetiolar node index: PPW/PW * 100

LPpI Lateral postpetiole index: PPL/PPH * 100

DPpI Dorsal postpetiole index: PPW/PPL * 100

PPI Postpetiole index: PPW/PTW * 100.

Figure 1 Schematic line drawings of Tetramorium jarawa sp. n. illustrating the measurements used in this study.

(A) Profile view with measuring lines for EL, WL, PH, PTH, PPH. (B) Mesosoma in dorsal view with measuring line for PW, (C) petiole and post-petiole in dorsal view with measuring lines for PTL, PTW, PPW, PPL, (D) head in full-face view with measuring lines for HL, HW, SL, (E) dorsocaudal view of the propodeum with measuring line for PSL.

Montage images and illustrations

Raw images of the new species were taken with a Leica DFC450 camera attached to a Leica M205C microscope and Leica Application Suite (version 4.1). The raw photo stacks were then processed to single montage images with Helicon Focus (version 6). Additional montage images used for the illustrated identification keys were taken from AntWeb (https://www.antweb.org). Vector illustrations were created with Adobe Illustrator (version CS 5) by tracing specimen photographs. All montage images used in this publication are available on AntWeb.

Micro X-ray computed tomography

Micro-CT scans were performed using a ZEISS Xradia 510 Versa 3D X-ray microscope and the ZEISS Scout and Scan Control System software (version 10.7.2936; Zeiss, Oberkochen, Germany). Specimen preparation and scanning protocol follows Hita Garcia et al. (2017a). For each species we scanned the holotype worker specimen. An overview of scanning settings is provided in Table 1. 3D reconstructions of the resulting scans were done with XMReconstructor (version 10.7.2936) and saved in DICOM file format (default settings; USHORT 16 bit output data type). Post-processing of DICOM raw data was performed with Amira software (version 6.1.1). The methodology for the virtual examinations of 3D surface models, generation of 3D rotation videos, and 3D PDFs also follows Hita Garcia et al. (2017a). Programs used for creating 3D PDFs are Meshlab (version 1.3.3) and Adobe Acrobat Pro DC (version 2015.006.30119) using the Tetra4D Converter plug-in (version 5.1.2). When viewing the 3D PDFs with Adobe Acrobat Reader (version 8 or higher), trusting the document by clicking on the image will activate the interactive 3D-mode and allows rotating, moving and zooming into the 3D model.

Table 1 Data summary of the two holotype specimens used for micro-CT scanning with an overview of specimen data, scan settings, and voxel sizes for the resulting scans (both holotypes are workers and all files are in DICOM format).

Species	Body part scanned	Specimen identifier	Voxel size (μm)	Exposure time (s)	Power (W)	Voltage (kV)	Amperage (μA)	
Tetramorium jarawa	Full body	NCBS-AV761	2.3852	1	5	60	83	
Tetramorium krishnani	Full body	NCBS-AV940	2.5343	1.5	4	50	80	

Data availability

All specimens used in this study have been databased and the data is freely accessible on AntWeb (http://www.antweb.org). Each specimen can be traced by a unique specimen identifier attached to its pin (e.g., NCBS-AV761). The Cybertype datasets provided in this study consist of the full micro-CT original volumetric datasets, 3D PDFs, 3D rotation video files, all light photography montage images, and all image plates including all important images of 3D models for each species. All data have been archived at Figshare (https://figshare.com/s/5594e5996963216c40cd) and are freely available. In addition to the cybertype data at Figshare, we also provide freely accessible 3D surface models of both holotypes on Sketchfab (https://sketchfab.com/arilab).

Nomenclatural acts

The electronic version of this article in Portable Document Format (PDF) will represent a published work according to the International Commission on Zoological Nomenclature (ICZN), and hence the new names contained in the electronic version are effectively published under that Code from the electronic edition alone. This published work and the nomenclatural acts it contains have been registered in ZooBank, the online registration system for the ICZN. The ZooBank LSIDs (Life Science Identifiers) can be resolved and the associated information viewed through any standard web browser by appending the LSID to the prefix http://zoobank.org/. The LSID for this publication is: urn:lsid:zoobank.org:pub:5943B1C2-8978-4ECB-AB48-8ADB0A89E30. The online version of this work is archived and available from the following digital repositories: PeerJ, PubMed Central and CLOCKSS.

Results

The Tetramorium fauna of India

The updated list of Indian Tetramorium species given below is based on Bharti (2011), Bharti & Kumar (2012), and Bharti et al. (2016) with some corrections and additions resulting from this study (Tables 2 and 3). Currently, we recognize 42 species for the country (Fig. 2), which belong to 12 species groups. We consider 27 species as endemic to India (Table 2), which translates to an endemism rate of 64%.

Table 2 Updated species list for the genus Tetramorium in India.

Species group	Species	Describers	Endemic	Exotic	Comments	
angulinode	smithi	Mayr, 1879				
bicarinatum	bicarinatum	(Nylander, 1846)				
bicarinatum	indicum	Forel, 1913				
bicarinatum	pacificum	Mayr, 1870				
bicarinatum	petiolatum	Sheela & Narendran, 1998	Yes			
bicarinatum	scabrum	Mayr, 1879				
caespitum	nursei	Bingham, 1903				
ciliatum	shivalikense	Bharti & Kumar, 2012	Yes			
fergusoni	fergusoni	Forel, 1902	Yes			
inglebyi	elisabethae	Forel, 1904	Yes			
inglebyi	inglebyi	Forel, 1902	Yes		Dubious records from Borneo Wang & Foster (2016), Zryanin (2011), and China Yunnan: Xu (1998), Qiao et al. (2009), Guénard & Dunn (2012)	
inglebyi	myops	Bolton, 1977	Yes			
inglebyi	triangulatum	Bharti & Kumar, 2012	Yes			
inglebyi	jarawa sp. n.		Yes			
melleum	mayri	(Forel, 1912)				
melleum	wroughtoni	(Forel, 1902)				
mixtum	malabarense	Sheela & Narendran, 1998	Yes			
mixtum	mixtum	Forel, 1902	Yes		Record from Borneo dubious Sukimin, Mohamed & Aris (2010)	
mixtum	rugigaster	Bolton, 1977	Yes			
mixtum	sentosum	Sheela & Narendran, 1998	Yes			
obesum	coonoorense	Forel, 1902	Yes			
obesum	decamerum	(Forel, 1902)	Yes			
obesum	lanuginosum	Mayr, 1870				
obesum	obesum	André, 1887				
obesum	rossi	(Bolton, 1976)	Yes			
simillimum	caldarium	(Roger, 1857)		Yes		
simillimum	simillimum	(Smith, 1851)		Yes		
tonganum	barryi	Mathew, 1981	Yes			
tonganum	christiei	Forel, 1902	Yes			
tonganum	salvatum	Forel, 1902	Yes			
tonganum	krishnani sp. n.	Yes				
tortuosum	belgaense	Forel, 1902	Yes			
tortuosum	keralense	Sheela & Narendran, 1998	Yes			
tortuosum	pilosum	Emery, 1893	Yes		Record from Zhejiang dubious Tang et al. (1985), Guénard & Dunn (2012)	
tortuosum	urbanii	Bolton, 1977	Yes			
tortuosum	tortuosum	Roger, 1863				
tortuosum	yerburyi	Forel, 1902	Yes		Record from Yunnan dubious Huang & Zhou (2007), Guénard & Dunn (2012)	
walshi	cordatum	Sheela & Narendran, 1998	Yes			
walshi	kheperra	(Bolton, 1976)				
walshi	walshi	(Forel, 1890)				
unclear	beesoni	(Mukerjee, 1934)	Yes		Initially described as Myrmica, then placed in Tetramorium by Radchenko & Elmes (2010); species group unknown.	
unclear	meghalayense	Bharti, 2011	Yes		In original description (Mathew & Tiwari, 2000) placed in Tetramorium bicarinatum group but based on the line drawings provided this placements is dubious	
Note:

For each species, we provide species group data, describer’s reference, classification as Indian endemic or exotic species, as well as comments on taxonomic status and distribution outside India.

Table 3 List of excluded Tetramorium species previously listed for India with arguments for exclusion decision.

Species group	Species	Describers	Comments	
tonganum	tonganum	Mayr, 1870	All the records of this species from India Bharti & Kumar (2012), Bharti et al. (2016) are based on a misidentification in Bharti & Kumar (2012). The species presented in that study is actually T. salvatum	
unclear	browni	Mathew & Tiwari, 2000	The name T. browni Mathew & Tiwari, 2000 was shown to be a junior primary homonym of T. browni Bolton, 1980 by Bharti (2011) who provided the replacement name T. meghalayense Bharti, 2011	

Figure 2 Tetramorium diversity in India.

(A) Overview of Tetramorium species richness in India. (B) Type localities of the new species on Havelock Island, part of Andaman Islands archipelago.

We note that the species list of Tetramorium given by Bharti et al. (2016) contains two erroneous species records (Table 3). The name T. browni Mathew & Tiwari, 2000 was shown to be a junior primary homonym of T. browni Bolton, 1980 by Bharti (2011) who provided the replacement name T. meghalayense Bharti, 2011. Consequently, we exclude Tetramorium browni from our species list. Another species we exclude from the Indian fauna is T. tonganum Mayr, 1870. This species was first reported from Uttar Pradesh and Himachal Pradesh by Bharti & Kumar (2012) and later by Bharti et al. (2016). After examining the description and images provided by Bharti & Kumar (2012), it is apparent that this record is based on a misidentification. The correct identity of the material listed as Tetramorium tonganum is actually T. salvatum Forel, 1902. In spite of these species being morphologically close, they differ in a variety of characters, most importantly the shape of the petiole. Tetramorium tonganum has a very long and curved peduncle, which strongly contrasts with the very short and straight peduncle of Tetramorium salvatum (illustrated in Tetramorium tonganum group identification key). Our finding is also strongly supported by the fact that the latter species is widespread in the montane and humid subtropical regions of Northeastern India and Pakistan, whereas Tetramorium tonganum is widely distributed throughout most of the Indomalayan and Australasian regions.

We encountered literature records from outside India of species considered by us as endemic to India. After examining the identification level and taxonomic expertise of these studies, we consider these records as highly suspicious and very likely misidentifications. We provide references for these in Table 2 under the respective misreported species.

Moreover, as most Asian countries, India has a small proportion of Tetramorium species that are not native. Bharti et al. (2016) list T. bicarinatum (Nylander, 1846), T. caldarium (Roger, 1857), T. pacificum Mayr, 1870, T. simillimum (Smith, 1851), and T. tonganum as exotics. In the cases of Tetramorium caldarium and Tetramorium simillimum we strongly agree with that assessment since these are certainly species of Afrotropical origin. The classification for the other three species is not that straightforward. As noted above, the record for Tetramorium tonganum was based on a misidentification with a native species, thus erroneous. Tetramorium bicarinatum is undoubtedly one of the most successful cosmopolitan tramps within the genus and among ants in general. Even though there is no hard evidence, most authors agree that its native range is likely somewhere in Southeast Asia (Bolton, 1977, 1979; Deyrup, Davis & Cover, 2000; Hita Garcia & Fisher, 2011; McGlynn, 1999). Consequently, without large-scale population genetic analyses it is impossible to infer if the species is introduced to India or a highly opportunistic and abundant member of the local fauna.

The classification of Tetramorium pacificum as exotic is problematic, too. One major problem is that the native range of this species is unknown. Schlick-Steiner, Steiner & Zettel (2006) opine that due to frequent human-mediated dispersal it might not be possible to ascertain the native range of this species, whereas other authors estimate its native range to be somewhere in the Indomalayan or Australasian regions including the archipelagos of the Pacific Ocean (Hita Garcia & Fisher, 2011; McGlynn, 1999). Furthermore, there is also a high degree of taxonomic uncertainty. Despite the fact that Tetramorium pacificum is easily recognizable in the Malagasy region, the Pacific, and the New World (Hita Garcia & Fisher, 2011), its identification in the Indomalayan region from India through South East Asia to New Guinea and Australia is very difficult. There are several native Southeast Asian species (T. scabrum Mayr, 1879 and T. manobo Calilung, 2000) that are sympatric with and morphologically almost indistinguishable from T. pacificum (Schlick-Steiner, Steiner & Zettel, 2006). Their identification requires considerable taxonomic skills and most publications providing records for any of these species, especially for India, need to be considered with extreme caution. To make matters worse, the taxonomy of the Tetramorium bicarinatum group in most of the regions in question is very much out of date due to unclear species delimitations and the existence of several potentially undescribed or cryptic species.

Identification key to Tetramorium species groups of India

As mentioned above, it is almost certain that future collecting in India will yield additional undescribed species. The latest species level key provided by Bharti & Kumar (2012) is essentially an updated key based on Bolton (1977) that included the species described since then. The key provided by Bharti & Kumar (2012) is a moderately good foundation for the identification of most of the currently known Indian species. However, numerous key couplets are based on rather weak and highly variable character states or absolute measurements (e.g., total body length or eye length size), which renders the key sometimes difficult to use for the existing species. Here we provide a newly developed and illustrated species group key that allows a straightforward placement of species into their respective species group, and can be used to supplement the existing species level keys. This is especially useful when dealing with undescribed species.

Species with distinctly branched (bifid, trifid, or very rarely quadrifid) hairs (Figs. 3A and 3B)2

– Species without branched hairs, hairs present neither bifid, trifid, nor quadrifid, either with simple pilosity (Fig. 3C), or with reduced pilosity but short appressed pubescence (Fig. 3D)3

Pilosity on first gastral tergite predominantly erect with hairs simple, bifid, or a combination of both (Fig. 3E)Tetramorium obesum group

– Pilosity on first gastral tergite predominantly suberect with trifid or very rarely quadrifid hairs (Fig. 3F)Tetramorium walshi group

Antennae 11-segmented (Fig. 3G)4

– Antennae 12-segmented (Fig. 3H)5

Antennal scrobes present and well-developed with margin all-around (Fig. 4A); antennal scapes shorter (SI 65–75); petiolar node in profile high rectangular nodiform with moderately rounded anterodorsal and posterodorsal angles (Fig. 4B)Tetramorium angulinode group

– Character combination never as above, especially antennal scrobes, if present, always much less conspicuous than above, and without well-developed posterior and ventral margins (Fig. 4C)Tetramorium tortuosum group

Head in full-face view distinctly cordate (Fig. 4D); lateral portion of clypeus modified into a low ridge in front of the antennal insertions (Fig. 4D); median cephalic and clypeal carinae/rugae absent (Fig. 4D)Tetramorium melleum group

– Head in full-face view never cordate as above (Fig. 4E); lateral portion of clypeus modified into a sharp and high ridge in front of the antennal insertions (Fig. 4E); median cephalic and clypeal carinae/rugae usually present, at least one of them (Fig. 4E)6

Sting appendage spatulate (Fig. 4F); frontal carinae very short, ending shortly behind level of posterior clypeal margin (Fig. 4G); antennal scrobes absentTetramorium fergusoni group

– Sting appendage triangular to dentiform, but never spatulate (Fig. 4H); frontal carinae usually conspicuous and much longer than above, rarely short or absent (Fig. 4I); antennal scrobes present or absent7

Base of first gastral tergite with anterodorsal angles projecting as a pair of blunt teeth or horns (Fig. 5A)8

– Base of first gastral tergite without anterodorsal angles projecting as a pair of blunt teeth or horns (Fig. 5B)9

Eyes moderately to strongly reduced, at most with five or six ommatidia in the longest row, usually just two or three. (Fig. 5C)Tetramorium inglebyi group

– Eyes moderately to well-developed and conspicuously much larger than above, at least with 10 or 11 ommatidia in the longest row (Fig. 5D)Tetramorium mixtum group

Frontal carinae very short to almost completely reduced and antennal scrobes absent (Fig. 5E)Tetramorium caespitum group

– Frontal carinae variably developed, but never reduced or absent as above (Fig. 5F)10

Hairs on mesosomal dorsum equal to or shorter than maximum antennal scape width and stout apically (Fig. 6A)Tetramorium simillimum group

– Hairs on mesosomal dorsum usually significantly longer than maximum antennal scape width, if short then fine and acute apically (Fig. 6B)11

Anterior clypeal margin with distinct median impression, small in some species but always distinct (Fig. 6C)Tetramorium bicarinatum group

– Anterior clypeal margin complete, without median impression (Fig. 6D)12

Propodeal spines comparatively longer; metatibiae with conspicuous suberect to erect hairs on outer surface (Fig. 6E)Tetramorium ciliatum group

– Propodeal armament variable, ranging from almost absent to short teeth/spines, but always shorter than above; metatibiae with (usually) appressed to (rarely) decumbent hairs on outer surface (Fig. 6F)Tetramorium tonganum group

Figure 3 Differences in pilosity on body, gastral tergite, and antennal segments.

Body in profile (A) T. rossi (CASTYPE12543), (B) T. obesum (CASENT0280874), (C) T. belgaense (CASENT0280882), (D) T. simillimum (CASENT0102390). First gastral tergite in profile (E) T. obesum (CASENT0280874), (F) T. walshi (CASENT0909098). Antennal funiculus, (G) T. smithi (CASENT0178421), (H) T. belgaense (CASENT0280882). Image (B, E, H) by Estella Ortega, image (F) by Zach Lieberman, image (G) by Erin Prado; all images from https://www.antweb.org.

Figure 4 Differences in head, petiole and sting appendage.

Lateral head (antennal scrobe within ellipses) and petiole in profile view (A, B) T. smithi (CASENT0178421), (C) T. pilosum (CASENT0280881). Head in full-face view (clypeus within ellipses) (D) T. wroughtonii (CASENT0909204), (E) T. fergusoni (CASENT0909167). Sting appendage and head in full-face view (frontal carinae within ellipses) (F) T. smithi (CASENT0790832), (G) T. fergusoni (CASENT0901104, CASENT0909167), (H, I) T. mixtum (CASENT0790833, CASENT0280896). Image (A) by Erin Prado, image (I) by Estella Ortega; all images (except F and H) from https://www.antweb.org.

Figure 5 Differences in gaster, eye size, and frontal carinae.

First gastral tergite in dorsal view (A) T. jarawa sp. n. (NCBS-AV761), (B) T. krishnani sp. n. (NCBS-AV940). Head in profile view (eyes under ellipses) (C) T. jarawa sp. n. (NCBS-AV761), (D) T. rugigaster (CASENT0901105). Head in full face view (frontal carinae in ellipses), (E) T. nursei (CASENT0901103), (F) T. krishnani sp. n. (NCBS-AV940). Images (D, E) by Ryan Perry; from https://www.antweb.org.

Figure 6 Differences in mesosoma & metatibiae pilosity and clypeus shape.

Mesosoma profile view showing pilosity on dorsum of mesosoma. (A) T. simillimum (CASENT0102390), (B) T. tonganum (CASENT0103250). Head in full-face view (clypeus in ellipses), (C) T. bicarinatum (CASENT0125127), (D) T. barryi (CASENT0280889). Pilosity on metatibiae (E) T. flagellatum (CASENT0901097), (F) T. tonganum (CASENT0171074). Images by April Nobile, Michele Esposito, Ryan Perry, and Eli Sarnat; from https://www.antweb.org.

Species group accounts

Tetramorium angulinode species group

Diagnosis: Eleven-segmented antennae; anterior clypeal margin notched and unspecialized; eyes of moderate size; antennal scapes short, not surpassing posterior head margin; antennal scrobes conspicuous and very well developed with clearly defined margins all-around; frontal carinae present and strongly developed; base of first gastral tergite not concave in dorsal view, without tubercles or teeth on each side; pilosity on dorsal surfaces of body erect with long and fine hairs; sting appendage spatulate (Fig. 7).

Figure 7 T. smithi (CASENT0909189), member of the T. angulinode group.

(A) Body in profile view, (B) body in dorsal view, (C) head in full-face view. Images by Zach Lieberman; from https://www.antweb.org.

Comments: This single Indomalayan member of this group represents a remarkable faunal oddity since it is the only member of a group of species otherwise endemic to the Afrotropical region. Interestingly, the distribution of the group is highly disjunctive with Tetramorium smithi being widely distributed in the Indomalayan and Australasian regions but strongly separated from its African relatives. Considering that the known distribution range of Tetramorium smithi grows consistently, and even reaches several pacific island archipelagos (Clouse, 2007), it appears that the species possesses some potential as tramp species. Nevertheless, based on some of our recent morphological examination of material from several South East Asian localities, there is also the possibility that Tetramorium smithi as currently understood represents a complex of rather cryptic species.

Tetramorium bicarinatum species group

Diagnosis: Twelve-segmented antennae; anterior clypeal margin notched and unspecialized; eyes moderately sized to large; antennal scapes short to moderately long, not surpassing posterior head margin; antennal scrobes usually present, but shallow and not clearly defined posteriorly and ventrally; frontal carinae always strongly developed and reaching posterior head level; base of first gastral tergite not concave in dorsal view, without tubercles or teeth on each side; pilosity on dorsal surfaces of body erect with long and fine hairs; sting appendage dentiform (Fig. 8).

Figure 8 T. indicum (CASENT0909109), member of the T. bicarinatum group.

(A) Body in profile view, (B) body in dorsal view, (C) head in full-face view. Images by Will Ericson; from https://www.antweb.org.

Comments: This is a relatively species-rich group with 16 species in the Indomalayan and Australasian regions and nine in the Afrotropical region. At present, the group is represented in India by five species, of which two are global tramps. As outlined above, it is challenging to ascertain if they are native members of the Indian myrmecofauna or introduced. The taxonomy of the group on the whole is complicated and identifications with the currently available resources are often challenging. In parts this is also true for the Indian species. The species delimitations of Tetramorium indicum, Tetramorium pacificum, and Tetramorium scabrum are not clear and misidentifications can occur easily. The identity of Tetramorium petiolatum is even more doubtful. Its original description is of comparatively poor quality and the authors state that the species is close to T. pacificum (Sheela & Narendran, 1998). However, based on their line drawings it looks very much like Tetramorium bicarinatum and Tetramorium indicum. Since the type material is not available for examination the true identity of this species will remain obscure.

Tetramorium caespitum species group

Diagnosis: Twelve-segmented antennae; anterior clypeal margin complete and unspecialized; eyes of moderate size; antennal scapes of moderate length, not surpassing posterior head margin; antennal scrobes absent; frontal carinae very short to almost completely reduced; base of first gastral tergite not concave in dorsal view, without tubercles or teeth on each side; pilosity on dorsal surfaces of body erect to suberect with long and stout hairs; sting appendage dentiform (Fig. 9).

Figure 9 T. nursei (CASENT0901103), member of the T. caespitum group.

(A) Body in profile view, (B) body in dorsal view, (C) head in full-face view. Images by Ryan Perry; from https://www.antweb.org.

Comments: As noted by Bolton (1977), Tetramorium caespitum group is the only endemic Tetramorium species group in the Palearctic, and it is widely distributed throughout all of Eurasia. Currently, it contains around 80 species and subspecies but this count has to be taken with a lot of caution and does not likely represent a realistic number. Compared to all other Tetramorium species groups, the Tetramorium caespitum group has never been comprehensively revised. Despite some recent small-scale revisionary treatments (Csösz, Radchenko & Schulz, 2007; Csösz & Schulz, 2010), its taxonomic situation can be classified as chaotic and no reliable identification resources exist. So far, only one species of the group is known from India: Tetramorium nursei. It occurs in Northwestern India representing the only genuine Palearctic component within the Indian Tetramorium fauna. It should be noted that the record of Tetramorium nursei from Kerala (Saranyan et al., 2013) is extremely dubious and very likely a misidentification. Species of the Tetramorium caespitum group are adapted to temperate and arid subtropical climate, thus not likely to occur in the Western Ghats.

Tetramorium ciliatum species group

Diagnosis: Twelve-segmented antennae; anterior clypeal margin complete and unspecialized; eyes of moderate size; antennal scapes short to moderately long, not surpassing posterior head margin; antennal scrobes extremely variable, ranging from completely absent to strongly developed with well-defined margin all-around; frontal carinae variably developed but always long and well surpassing eye level; base of first gastral tergite not concave in dorsal view, without tubercles or teeth on each side; pilosity on dorsal surfaces of body erect with long to extremely long and fine hairs; sting appendage dentiform (Fig. 10).

Figure 10 Tetramorium shivalikense, member of the Tetramorium ciliatum group, (A) head in full-face view, (B) body in profile view, (C) body in dorsal view.

Images reproduced from Bharti & Kumar (2012).

Comments: When proposing this group Bolton (1977) pointed out that he saw it more as a convenience group for species that did not fit well in other, better delineated species groups. Now, 40 years after Bolton’s (1977) revision, the situation has not changed and the group still represents an amalgamation of species with diverging morphological character sets. Of the nine valid species of the group, eight are distributed in South East Asia and only one species occurs in India: Tetramorium shivalikense. Whether the latter is indeed related to the other members of the group and if the group as whole is monophyletic or polyphyletic remains unclear. This can only be resolved with a comprehensive large-scale phylogenetic analysis including all Indomalayan species groups in general and most species of the Tetramorium ciliatum group in particular.

Tetramorium fergusoni species group

Diagnosis: Twelve-segmented antennae; anterior clypeal margin complete and median portion with narrow but distinct lamelliform apron; eyes of moderate size; antennal scapes short, not surpassing posterior head margin; antennal scrobes absent; frontal carinae very short, almost absent; propodeum with very long spines; base of first gastral tergite not concave in dorsal view, without tubercles or teeth on each side; pilosity on dorsal surfaces of body predominantly erect with relatively short and thick hairs, some hairs flattened; sting appendage spatulate (Fig. 11).

Figure 11 T. fergusoni (CASENT0901104), member of the T. fergusoni group.

(A) Body in profile view, (B) body in dorsal view, (C) head in full-face view. Images by Ryan Perry; from https://www.antweb.org.

Comments: The Tetramorium fergusoni group is a monotypic group endemic to India with an interesting character set. The possession of 12-segmented antennae, a modified anterior clypeal margin, and a spatulate sting appendage is a unique combination not seen in another Tetramorium on a global scale. Due to its rather unusual morphology, it’s not possible to ascertain the affinities of this species to other Tetramorium groups.

Tetramorium inglebyi species group

Diagnosis: Twelve-segmented antennae; anterior clypeal margin variable; eyes very small to minute; antennal scapes short, not surpassing posterior head margin; antennal scrobes absent; frontal carinae either completely absent or strongly reduced, at most reaching eye level; base of first gastral tergite strongly concave in dorsal view, the anterolateral corners produced as short tubercle or tooth on each side; pilosity on dorsal surfaces of body erect; sting appendage dentiform (Fig. 12).

Figure 12 T. inglebyi (CASENT0280897), member of the T. inglebyi group.

(A) Body in profile view, (B) body in dorsal view, (C) head in full-face view. Images by Estella Ortega; from https://www.antweb.org.

Comments: As already pointed out by Bolton (1977), this is a small group of relatively rare ants endemic to India. The key characters that define it are the comparatively small eyes and the shape of the base of the first gastral tergite. So far, only five species are known and each only from few specimens and collections. There is no available knowledge on the biology of the group. However, the head shape and the small eyes are reminiscent of the Afrotropical Tetramorium shilohense group. Some members of the latter group are known to be termitophagous, and it could be possible that the species of the Tetramorium inglebyi share that dietary adaptation. However, this is highly speculative and needs to be confirmed through field observations.

Identification key to Indian species of the Tetramorium inglebyi group (workers)

Eyes large, always longer than maximum width of antennal scapes (Fig. 13A)Tetramorium inglebyi

– Eyes much smaller than above, always shorter than maximum width of antennal scapes (Figs. 13B–13E)2

Propodeal spines relatively shorter (Fig. 13F); dorsum of petiolar node in dorsal view conspicuously much broader than long (Fig. 13H); procoxae never completely reticulate–punctate3

– Propodeal spines relatively longer (Fig. 13G); dorsum of petiolar node in dorsal view about as long as broad or clearly longer than broad (Fig. 13I); procoxae completely reticulate–punctate4

In profile petiolar node appearing higher and thinner (Fig. 14A); dorsum of promesonotum reticulate–rugose (Fig. 14C); first gastral tergite unsculptured, smooth and shinyTetramorium elisabethae

– In profile petiolar node appearing thicker and more compact (Fig. 14B); dorsum of promesonotum longitudinally rugulose (Fig. 14D); base of first gastral tergite longitudinally ruguloseTetramorium triangulatum

Propodeum with comparatively longer spines (Fig. 14E); in profile peduncle of petiole with large anteroventral lamella; shape of petiolar node narrowing from base to apex and dorsum convex (Fig. 14E)Tetramorium myops

– Propodeum with comparatively shorter spines (Fig. 14F); in profile peduncle of petiole without large anteroventral lamella; shape of petiolar node appearing square and dorsum straight (Fig. 14F)Tetramorium jarawa sp. n.

Figure 13 Differences in eye size, propodeal spines, and petiole shape.

Head in full-face view (eyes within ellipses) (A) T. inglebyi (CASENT0280897), (B) T. elisabethae (CASENT0909166), (C) T. inglebyi (CASENT0280897), (D) T. triangulatum, (E) T. jarawa sp. n. (NCBS-AV761). Mesosoma in profile view (F, G) and dorsum of waist segments (H, I). Arrows indicate propodeal spines/teeth and petiole. (F) T. elisabethae (CASENT0901107), (G) T. jarawa sp. n. (NCBS-AV761), (H) T. elisabethae (CASENT0901107), (I) T. jarawa sp. n. (NCBS-AV761). Images (except E, G and I) by Estella Ortega, Will Ericson, and Zach Lieberman; from https://www.antweb.org.

Figure 14 Differences in petiole shape, sculpturing on mesosoma, and propodeal spines.

Petiole within ellipses in profile view (A, B) and arrows indicating sculpturation on dorsum of mesosoma (C, D). (A, C) T. elisabethae (CASENT0901107), (B, D) T. triangulatum (reproduced from Bharti & Kumar, 2012). Petiole (lamella in ellipse) and propodeal spines (indicated by arrows) in profile view. (E) T. myops (CASENT0901106), (F) T. jarawa sp. n. (NCBS-AV761). Images (A, C, E) by Zach Lieberman and Ryan Perry; from https://www.antweb.org. Images (B, D) from Bharti & Kumar (2012).

Tetramorium jarawa sp. n.

Type material

Holotype, pinned worker, INDIA, Andaman Islands archipelago, Havelock Island, 11.975817 N, 93.016897 E, 5 m, tropical (semi) evergreen forest, sifted leaf-litter, 26.XI.2015 (NCBS: NCBS-AV761).

Paratype, one pinned worker with same data as holotype (NCBS: NCBS-AV931).

Cybertype, volumetric raw data (in DICOM format), 3D PDF, and 3D rotation video of the physical holotype (NCBS: NCBS-AV761) in addition to montage photos illustrating head in full-face view, profile and dorsal views of the body of both specimens. The data is deposited in Figshare (https://figshare.com/s/5594e5996963216c40cd) and can be freely accessed as virtual representations of the types. In addition to the cybertype data at Figshare, we also provide a freely accessible 3D surface model of the holotype at Sketchfab (https://sketchfab.com/models/e3283a5fa4134c9c84ee953b357796c1).

Diagnosis: The following character combination distinguishes Tetramorium jarawa from the remainder of the Tetramorium inglebyi group: very small eyes (OI 9–10); relatively short scape (SI 60–71); propodeal spines moderately long (PSLI 18); peduncle of petiole without large anteroventral lamella; in profile petiolar node appearing square and dorsum flat; in dorsal view dorsum of petiolar node clearly longer than broad (Figs. 15 and 16).

Figure 15 Tetramorium jarawa sp. n. (NCBS-AV761, holotype).

(A) Body in profile view, (B) body in dorsal view, (C) head in full-face view.

Figure 16 3D surface model of Tetramorium jarawa sp. n. holotype worker (NCBS-AV761).

(A) Body in profile view, (B) body in dorsal view, (C) head in full-face view, (D) anterior head in anterofrontal view, (E) head in profile view, (F) first gastral tergite in dorsal view.

Worker measurements (N = 2): HL 0.56–0.59 (0.56); HW 0.52–0.54 (0.52); SL 0.32–0.38 (0.32); EL 0.05 (0.05); PH 0.27–0.28 (0.27); PW 0.35–0.37 (0.35); WL 0.62–0.65 (0.62); PSL 0.1 (0.1); PTL 0.14–0.16 (0.14); PTH 0.17–0.19 (0.17); PTW 0.16–0.17 (0.16); PPL 0.17 (0.17); PPH 0.18–0.19 (0.18); PPW 0.20–0.22 (0.20); CI 92–94 (94); SI 60–71 (60); OI 9–10 (10); DMI 56 (56); LMI 43–44 (44); PSLI 18 (18); PeNI 45 (45); LPeI 81–83 (81); DPeI 105–112 (112); PpNI 57–59 (57); LPpI 87–91 (91); DPpI 120–130 (120); PPI 126–130 (126).

Worker description: Head longer than wide (CI 92–94); posterior head margin weakly concave. Anterior clypeal margin complete and convex. Frontal carinae very weakly developed to absent; antennal scrobes absent. Antennal scapes short, not surpassing posterior head margin (SI 60–71). Eyes very small (OI 9–10), composed of 2–3 facets in longest row. Mesosomal outline in profile weakly convex to flat, moderately marginate from lateral to dorsal mesosoma; promesonotal suture and metanotal groove absent. Propodeum armed with moderately long spines (PSLI 18), their tips slightly curved upwards. Propodeal lobes well-developed and triangular. Petiolar node nodiform, appearing square, slightly higher than long (LPeI 81–83), anterior and posterior faces approximately parallel, anterodorsal and posterodorsal margins situated at about same height and both moderately rounded, petiolar dorsum flat to moderately convex; whole node in dorsal view about as long as wide (DPeI 105–112) (dorsum of node conspicuously longer than broad), in dorsal view pronotum approximately 2.1 times wider than petiolar node (PeNI 45). Postpetiole in profile globular, approximately 1.1 times higher than long (LPpI 86–87); in dorsal view around 1.2–1.3 times wider than long (DPpI 120–130), pronotum around 1.7–1.8 times wider than postpetiole (PpNI 57–59). Postpetiole in dorsal view around 1.3 times wider than petiolar node (PPI 126–127). Mandibles striate; clypeus longitudinally rugose/rugulose with well-developed median ruga; most of head strongly reticulate–rugose except for irregularly longitudinally rugose anterior cephalic dorsum close to posterior clypeal margin. Mesosoma laterally anteriorly irregularly rugose becoming more reticulate–punctate toward propodeum; dorsal mesosoma reticulate–rugose; forecoxae reticulate–punctate. Petiole and postpetiole laterally irregularly rugulose, dorsally smooth and shining. First gastral tergite unsculptured, smooth, and shiny. Ground sculpture very weak to absent on most of body. Dorsal surfaces of mesosoma, petiole, postpetiole with short to moderately long, thin, and apically sharp pilosity; dorsum of head with short hairs curved inward, somewhat decumbent. Anterior edges of antennal scapes and dorsal (outer) surfaces of hind tibiae with decumbent to suberect hairs. Mesosoma, head, petiole, and postpetiole dark reddish brown but head slightly lighter; mandibles, antennae, gaster, and legs brownish yellow.

Etymology: The species is named after the Jarawas, an indigeneous people from the Andaman Islands. The name is a noun in apposition and thus invariant.

Distribution and biology: Tetramorium jarawa is currently only known from its type locality on Havelock Island, in the Andaman Islands archipelago. Given the relatively small size of the island and its proximity to one of the bigger islands of the archipelago, it may be speculated that the species will be present on other islands of the archipelago as well. Tetramorium jarawa was collected from leaf-litter in an undisturbed patch of evergreen forest. In the island-wide ant diversity survey done using Winkler transects (80 m length, 5 × 1 m2 leaf-litter collected in each transect), the species was found only in one out of 22 transects in evergreen forests. It thus appears to be rare and restricted to the inland evergreen forests, as it was not found in coastal forests and other disturbed habitats of the island despite considerable sampling effort. If, as mentioned above, this species has indeed a termitophagous, cryptic lifestyle in close proximity to termites, this would explain its rarity in collections.

Diagnostic comments: This new species is straightforwardly recognizable with the diagnosis and key provided above. There is no doubt that Tetramorium jarawa is a member of the Tetramorium inglebyi group, but its relationships to the other four members are unclear. Due to the scarce material available for this study, it is not possible to ascertain any levels of intraspecific variation.

Our morphometric description of the petiolar node should be taken with caution. We used the standard measurement for petiolar width (PTW) and length (PTL) used in most previous studies on Tetramorium taxonomy (Hita Garcia, Fischer & Peters, 2010; Hita Garcia & Fischer, 2014; Bharti & Kumar, 2012) to calculate DPeI. This index is supposed to provide a measure for how broad the node appears in dorsal view for a majority of species within the genus and has proven successful for more than 150 species treated in previous studies. However, in cases where the dorsum of the node is less broad than the base this leads to biased results. The DPeI of 105–112 generated for Tetramorium jarawa gives the impression that the node is weakly broader than long, but the dorsum of the node is obviously much longer than broad.

Tetramorium melleum species group

Diagnosis: Twelve-segmented antennae; head shape conspicuously cordate, narrowing anteriorly and broadening posteriorly with strongly concave posterior head margin; anteromedian margin of clypeus arcuate to triangular and conspicuously projecting over mandibles; eyes of moderate size; antennal scapes moderately long; antennal scrobes absent; frontal carinae very short to literally absent; base of first gastral tergite not concave in dorsal view, without tubercles or teeth on each side; pilosity on dorsal surfaces of body erect with long and fine hairs; sting appendage dentiform (Fig. 17).

Figure 17 T. wroughtonii (CASENT0909204), member of the T. melleum group.

(A) Body in profile view, (B) body in dorsal view, (C) head in full-face view. Images by Zach Lieberman; from https://www.antweb.org.

Comments: Due to the very distinctive head modifications, the species of this group were placed, until very recently, together with their Afrotropical relatives in their own genus: Rhoptromyrmex Mayr, 1901. However, a recent molecular study of the subfamily Myrmicinae provided evidence that Rhoptromyrmex is nested within Tetramorium and consequently not monophyletic (Ward et al., 2015). In spite of the worker and male castes of this group being very similar among species, the queens display an incredible phenotypical variation. This morphological variability is related to divergent lifestyles since it is known that the members of this group demonstrate various stages of social parasitism, ranging from autoparasites through temporary social parasites to workerless inquilines (Bolton, 1976, 1986; Brown, 1964).

At present, only two valid species are known from the region, of which one is found in India. However, on the basis of morphological examinations of material from throughout the whole Indomalayan region, one can observe an astonishing intraspecific variation within Tetramorium wroughtonii, and it is very likely that this species is actually a complex of at least 10 more or less cryptic species.

Tetramorium mixtum species group

Diagnosis: Twelve-segmented antennae; anterior clypeal margin unspecialized, usually entire, rarely notched; eyes of moderate size; antennal scapes short, not surpassing posterior head margin; antennal scrobes variably developed; frontal carinae well-developed and surpassing eye level; base of first gastral tergite not concave in dorsal view, without tubercles or teeth on each side; pilosity on dorsal surfaces of body erect with long and fine hairs; sting appendage dentiform (Fig. 18).

Figure 18 T. rugigaster (CASENT0901105), member of the T. mixtum group.

(A) Body in profile view, (B) body in dorsal view, (C) head in full-face view. Images by Zach Lieberman; from https://www.antweb.org.

Comments: This is a small group with seven species occurring in India, Sri Lanka, and Vietnam. The group was initially proposed by Bolton (1977) to include the species with a strongly concave base of the first gastral tergite that do not belong to the Tetramorium inglebyi group. Its validity was questioned by Roncin (2002) who while describing two new species of the group from Vietnam opined that the Tetramorium mixtum group is artificial and its members should be placed in different groups. Without going into detail, we admit that Roncin (2002) was likely correct in doubting the Tetramorium mixtum group as a whole. However, it is quite probable that the four Indian and the Sri Lankan species form a natural group based on their shared morphology, whereas the species from Vietnam might belong to another Indomalayan group. Consequently, we prefer to maintain the Tetramorium mixtum group and defer any decision of its validity until a molecular phylogenetic analysis is conducted on this group.

Tetramorium obesum species group

Diagnosis: Antennae with 10 or 12 segments; anterior clypeal margin variable, complete or notched, but always unspecialized; eyes of moderate size; antennal scapes usually short to moderately long, not surpassing posterior head margin; antennal scrobes variably developed, from fully absent to strongly developed, deep, and with sharp margins all-around; frontal carinae weakly to strongly developed but always surpassing eye level; base of first gastral tergite not concave in dorsal view, without tubercles or teeth on each side; pilosity on dorsal surfaces of body either completely or partly branched, on first gastral tergite usually a mixture of simple and bifid; sting appendage dentiform (Fig. 19).

Figure 19 T. rossi (CASENT0901054), member of the T. obesum group.

(A) Body in profile view, (B) body in dorsal view, (C) head in full-face view. Images by Will Ericson; from https://www.antweb.org.

Comments: The Tetramorium obesum group is distributed in the Indomalayan and Australasian regions, and with currently 11 described species of moderate size. Bolton (1976) recognized two species complexes on the basis of diverging morphology, one of which consists entirely of the five valid species from India.

Tetramorium simillimum species group

Diagnosis: Twelve-segmented antennae; anterior clypeal margin complete and unspecialized; eyes of moderate size; antennal scapes short to moderate, not surpassing posterior head margin; antennal scrobes either present without clear demarcation or absent; frontal carinae moderately to strongly developed but always surpassing eye level; base of first gastral tergite not concave in dorsal view, without tubercles or teeth on each side; pilosity on dorsal surfaces of body erect with short and thick hairs; sting appendage dentiform (Fig. 20).

Figure 20 T. simillimum (CASENT0919927), member of the T. simillimum group.

(A) Body in profile view, (B) body in dorsal view, (C) head in full-face view. Images by Michele Esposito; from https://www.antweb.org.

[Note: this diagnosis is only applicable to the few species occurring in the Indomalayan region, and not to the remainder of the Afrotropical group fauna].

Comments: This is one of the larger species groups within Tetramorium with approximately 30 species, most of which are endemic to the Afrotropical region. Two members of the group have become extremely successful panglobal tramps: Tetramorium caldarium and Tetramorium simillimum. Both are found in all zoogeographic regions, and, not surprisingly, also in India.

Tetramorium tonganum species group

Diagnosis: Twelve-segmented antennae; eyes moderately sized to large; frontal carinae reaching beyond the level of the posterior eye margins but usually weakly developed; base of first gastral tergite not concave in dorsal view and anterolateral corners not produced as short tubercles or teeth; pilosity on dorsal surfaces of body erect; antennal scapes and dorsal surfaces of hind tibiae only with short subdecumbent to appressed pubescence; sting appendage dentiform (Fig. 21).

Figure 21 T. barryi (CASENT0280889), member of the T. tonganum group.

(A) Body in profile view, (B) body in dorsal view, (C) head in full-face view. Images by Michele Esposito; from https://www.antweb.org.

Comments: As pointed out by Bolton (1977), this is a widespread group of species in the Indomalayan and Australasian regions. While most of the individual species are known to have small ranges, Tetramorium tonganum is distributed widely and also recorded from hothouses in the temperate regions. Currently, 11 valid species are recognized, of which we recognize four to occur in India. The species in this group are morphologically very close to the members of the Tetramorium scabrosum group. Basically, the lack of long, standing hairs on the outer margins of the legs is the only character separating both groups (Bolton, 1977).

Identification key to Indian species of the Tetramorium tonganum group (workers)

As outlined above, we exclude Tetramorium tonganum from the currently known Indian Tetramorium fauna. However, since the species is very widespread in tropical Asia and already recorded from Sri Lanka, it is possible that it occurs in the humid, tropical regions of India but has not been collected yet. Moreover, there is a possibility of human-mediated introduction of the species into India in the future. Consequently, in order to facilitate future identifications, we include it in the species key below.

Propodeum unarmed without teeth or spines (Fig. 22A)Tetramorium krishnani sp. n.

– Propodeum armed with teeth or spines (Fig. 22B)2

In profile petiolar node appearing enlarged and conspicuously elongated and convex (Fig. 22C)Tetramorium barryi

– In profile petiolar node not appearing enlarged and significantly less elongated and convex (Figs. 22D–22F)3

In profile petiolar node low and appearing longer than high (Fig. 22D)Tetramorium christiei

– In profile petiolar node clearly higher and appearing higher than long (Figs. 22E and 22F)4

In profile peduncle of petiole long and curved (Fig. 22E)Tetramorium tonganum

– In profile peduncle of petiole short and straight, (Fig. 22F)Tetramorium salvatum

Figure 22 Differences in mesosomal armament and petiole shape.

Mesosoma in profile view. (A) T. krishnani sp. n. (NCBS-AV940) (B) T. tonganum (CASENT0103250). Petiole in profile view. (C) T. barryi (CASENT0280889), (D) T. christiei (CASENT0901088), (E) T. tonganum (CASENT0171074), (F) T. salvatum (CASENT0909173). Images (except A and B) by April Nobile, Michele Esposito, Ryan Perry, Eli Sarnat, and Zach Lieberman; from https://www.antweb.org.

Tetramorium krishnani sp. n.

Type material

Holotype, pinned worker, INDIA, Andaman Islands archipelago, Havelock Island, 12.003499 N, 92.993196 E, 93 m, tropical (semi) evergreen forest, sifted leaf-litter, 20.XI.2015 (NCBS: NCBS-AV940).

Paratypes, eight pinned workers: INDIA, Andaman Islands archipelago, Havelock Island: 12.0027333 N, 92.9396667 E, 32 m, tropical (semi) evergreen forest, sifted leaf-litter, 10.XI.2015 (NCBS: NCBS-AV755, NCBS-AV941, NCBS-AV942). 12.0038 N, 93.0041833 E, 42 m, leaf-litter, 23.XI.2015 (NCBS: NCBS-AV756, NCBS-AV757). 12.000607 N, 92.946047 E, 62 m, tropical (semi) evergreen forest, sifted leaf-litter, 19.XI.2015 (NCBS: NCBS-AV843). 11.961389 N, 92.991111 E, 38 m, tropical (semi) evergreen forest, sifted leaf-litter, 5.I.2016 (ZSI: ZSI-WGRC-IR-INV-9780, ZSI-WGRC-IR-INV-9781).

Cybertype, volumetric raw data (in DICOM format), 3D PDF, and 3D rotation video of the physical holotype (NCBS: NCBS-AV940) in addition to montage photos illustrating head in full-face view, profile and dorsal views of the body of both specimens. The data is deposited in Figshare (https://figshare.com/s/5594e5996963216c40cd) and can be freely accessed as virtual representations of the types. In addition to the cybertype data at Figshare, we also provide a freely accessible 3D surface model of the holotype at Sketchfab (https://sketchfab.com/models/298bc42063ad4deea36c4043427929f9).

Diagnosis: The following character combination distinguishes Tetramorium krishnani from the other species of the Tetramorium tonganum species group: propodeum fully unarmed without teeth or spines; petiole with very long and curved peduncle and low and elongated node (Figs. 23 and 24).

Figure 23 Tetramorium krishnani sp. n. (NCBS-AV940, holotype).

(A) Body in profile view, (B) body in dorsal view, (C) head in full-face view.

Figure 24 Still images from 3D surface model of Tetramorium krishnani sp. n. holotype worker (NCBS-AV940).

(A) Body in profile view, (B) body in dorsal view, (C) head in full-face view, (D) anterior head in anterofrontal view, (E) head in profile view, (F) first gastral tergite in dorsal view.

Worker measurements (N = 6): HL 0.58–0.64 (0.61); HW 0.51–0.57 (0.54); SL 0.40–0.46 (0.43); EL 0.12–0.14 (0.14); PH 0.28–0.33 (0.28); PW 0.37–0.42 (0.40); WL 0.67–0.78 (0.74); PSL 0.02 (0.02); PTL 0.22–0.26 (0.22); PTH 0.19–0.22 (0.20); PTW 0.18–0.20 (0.19); PPL 0.22–0.25 (0.25); PPH 0.17–0.19 (0.17); PPW 0.18–0.20 (0.20); CI 86–88 (88); SI 79–81 (80); OI 25–26 (26); DMI 52–56 (54); LMI 38–45 (38); PSLI 3.1–3.4 (3.2); PeNI 46–51 (47); LPeI 112–119 (112); DPeI 76–85 (84); PpNI 47–54 (49); LPpI 118–143 (143); DPpI 78–89 (78); PPI 100–104 (103).

Worker description: Head longer than wide (CI 86–88); posterior head margin very weakly concave. Anterior clypeal margin complete and convex. Frontal carinae well-developed, approaching corners of posterior head margin; antennal scrobes absent. Antennal scapes long, reaching posterior head margin (SI 79–81). Eyes large (OI 25–26). Mesosomal outline in profile moderately convex, very weakly marginate from lateral to dorsal mesosoma; promesonotal suture and metanotal groove absent; mesosoma comparatively stout and high (LMI 38–45). Propodeum unarmed. Propodeal lobes well-developed, triangular, with sharp tips. Petiolar node in profile view oval-shaped with rounded posterodorsal margin and almost flat anterodorsum margin, slightly longer than high (LPeI 112–119), anterodorsal margin situated slightly lower than posterodorsal margin, dorsum moderately convex; node in dorsal view 1.2–1.3 times longer than wide (DPeI 76–85), in dorsal view pronotum approximately 2–2.1 times wider than petiolar node (PeNI 46–51). Postpetiole in profile moderately convex, approximately 1.2–1.3 times longer than high (LPpI 118–143); in dorsal view around 1.2 times longer than wide (DPpI 78–89), pronotum around 2.0–2.1 times wider than postpetiole (PpNI 47–54). Postpetiole in dorsal view about as wide as petiolar node (PPI 100–104). Mandibles strongly striate; clypeus longitudinally rugose/rugulose, with 4–5 rugae/rugulae with well-developed median ruga; cephalic dorsum between frontal carinae reticulate–rugose to longitudinally rugose, posteriorly more reticulate–rugose and anteriorly more longitudinally rugose; lateral and ventral head mostly reticulate–rugose. Mesosoma dorsally and laterally reticulate–rugose to longitudinally rugose, dorsally more longitudinally rugose. Forecoxae with very weakly developed longitudinal rugae/rugulae. Petiole and postpetiole with very few (2–3), weakly developed longitudinal ruga/rugulae; their dorsa smooth and shining. First gastral tergite unsculptured, smooth, and shiny. Whole body sparsely covered with short to long fine standing hairs. Anterior edges of antennal scapes with decumbent to suberect hairs. Mesosoma, head, petiole, postpetiole, and gaster orangish brown; mandibles and legs lighter in color.

Etymology: The name of the new species is a patronym in honor of late Dr. K. S. Krishnan (Prof. Emeritus, NCBS) in appreciation of his scientific achievements, and his unbounded enthusiasm for—and support of—curiosity-driven ecological and wildlife research.

Distribution and biology: Tetramorium krishnani is recorded so far only from its type locality on Havelock Island, in the Andaman Islands archipelago but it is likely to be present on other islands of the archipelago. We recorded the species in 8 out of 35 total leaf-litter transects sampled on the island, suggesting that it was not rare. It appears to be restricted to the relatively undisturbed inland wet evergreen forests and only once was it found in the relatively drier coastal forests. Other aspects of its biology such as diet, colony structure, and behavior remain to be recorded.

Diagnostic comments: Tetramorium krishnani is easily distinguishable from the other species due to its very slender gestalt in addition to the very long peduncle, the complete lack of propodeal armament, and the relatively low and long node in lateral view. The type series is relatively small, thus does not allow any assessment of intraspecific variation.

Tetramorium tortuosum species group

Diagnosis: Eleven-segmented antennae; anterior clypeal margin notched and unspecialized; eyes of moderate size; antennal scapes variable, ranging from short to long; antennal scrobes variable, ranging from very reduced to strongly developed, in the latter case without clear margin all-around; frontal carinae always strongly developed; base of first gastral tergite not concave in dorsal view, without tubercles or teeth on each side; pilosity on dorsal surfaces of body erect, usually with long and fine, rarely short and thick, hairs; sting appendage spatulate (Fig. 25).

Figure 25 T. belgaense (CASENT0280882), member of the T. tortuosum group.

(A) Body in profile view, (B) body in dorsal view, (C) head in full-face view. Images from https://www.antweb.org.

Comments: The Tetramorium tortuosum species group is the most species-rich group of the genus with more than 50 described species, of which six are known from India. It is also the most widespread with native faunas in the Afrotropical, Malagasy, Indomalayan, and Australasian regions, as well as the New World. There are some considerable doubts if all of these represent indeed a monophyletic clade, and it is probable that the morphological similarities upon which they were placed in the same group are based on convergent evolution (Hita Garcia & Fisher, 2012a, 2013).

Tetramorium walshi species group

Diagnosis: Twelve-segmented antennae; anterior clypeal margin notched and unspecialized; eyes of moderate size; antennal scapes usually short, not surpassing posterior head margin; antennal scrobes always present and usually (at least in the Indian species) strongly developed, deep, and with sharply defined margin all-around; frontal carinae well-developed; base of first gastral tergite not concave in dorsal view, without tubercles or teeth on each side; pilosity on dorsal surfaces of body either completely or partly branched, all hairs on first gastral tergite trifid or quadrifid; sting appendage dentiform (Fig. 26).

Figure 26 T. walshi (CASENT0909092), member of the T. walshi group.

(A) Body in profile view, (B) body in dorsal view, (C) head in full-face view. Images by Zach Lieberman; from https://www.antweb.org.

Comments: This is another group of moderate size distributed throughout the whole Indomalayan and Australasian regions. Of the 15 valid species known so far, three are found in India.

Discussion

With 42 known species the Tetramorium fauna of India is the largest in Asia. As discussed above, the Indian Subcontinent is largely undersampled and we expect that species count to increase significantly with future collections. The Indian Tetramorium fauna also appears to be highly unique with an endemism rate of 64%. Based on unpublished data, this rate is only higher in sub-Saharan Africa, Madagascar, New Guinea, and Australia, all of which have rates of 90% or higher.

The distinctive character of the Indian fauna can also be observed in the species groups present. Most groups, such as the Tetramorium bicarinatum, Tetramorium melleum, Tetramorium tonganum, Tetramorium tortuosum, and Tetramorium walshi groups are widespread in the Indomalayan and Australasian regions and also well presented in India. The predominantly South East Asian Tetramorium ciliatum group and the Palearctic Tetramorium caespitum group have their main distributions elsewhere but extend into India where they are represented by one species each. The Tetramorium simillimum group contains two panglobal tramps of African origin. Finally, there are four “genuinely” Indian groups: the Tetramorium inglebyi, Tetramorium fergusoni, Tetramorium mixtum, and Tetramorium obesum groups. The Tetramorium inglebyi and Tetramorium fergusoni groups are entirely endemic to the Indian Subcontinent, whereas the Tetramorium mixtum and Tetramorium obesum groups (the Tetramorium obesum complex) have their main diversity in India but also have species further east in Southeast Asia. The faunal composition of these groups show that the Indian fauna is well embedded within the Indomalayan region but also has faunal affinities with the Palearctic, and most importantly, a high level of unique diversity.

Our work provides an update to the taxonomy of Indian Tetramorium fauna. It also highlights the fact that while certain groups of insects such as butterflies are fairly well documented in India (Kunte, Sondhi & Roy, 2017), the ant fauna still remains poorly documented and understood. For example, given that northeast India is known to have high biodiversity in many groups, but is also among the most undersampled regions in India, we anticipate that Tetramorium richness will increase several-fold there after further inventory efforts. Similarly, while the Western Ghats appears to be one of the most Tetramorium-rich regions in India, it is highly likely to be undersampled. Our thorough survey of leaf-litter ant communities on the small Havelock Island has yielded around a half-dozen undescribed ant species. Most of these species are likely to be endemic to the archipelago. Our survey underscores the need to extensively sample the ant fauna across India to better document and discover Indian ant diversity. Such surveys are expected to bring about advances in taxonomy and biodiversity documentation, which are critical for our general understanding of factors shaping ant biodiversity.

We thank the Department of Environment and Forest, Andaman and Nicobar Administration, for granting research and collection permits. We thank Ajit Kumar for his invaluable assistance during the fieldwork. Thanks are due to T.L. Audisio and K. Dudley from OIST for generating the line drawings and maps, respectively. Thanks to A. Jayadevan for help in proofreading the manuscript. We thank Dr. P. Sureshan of Zoological Survey of India (ZSI)—Calicut for help during deposition of type material in ZSI. GA is grateful to D. Agashe for mentorship and support.

Additional Information and Declarations

Competing Interests

Author Contributions

Field Study Permissions

Data Availability

New Species Registration

The authors declare that they have no competing interests.

Gaurav Agavekar conceived and designed the experiments, performed the experiments, analyzed the data, contributed reagents/materials/analysis tools, wrote the paper, prepared figures and/or tables, reviewed drafts of the paper.

Francisco Hita Garcia conceived and designed the experiments, analyzed the data, wrote the paper, prepared figures and/or tables, reviewed drafts of the paper.

Evan P. Economo conceived and designed the experiments, contributed reagents/materials/analysis tools, wrote the paper, prepared figures and/or tables, reviewed drafts of the paper.

The following information was supplied relating to field study approvals (i.e., approving body and any reference numbers):

Field research and collection permits were provided by the Department of Environment and Forest, Andaman and Nicobar Administration (Permit no. CWLW/WL/134/353).

The following information was supplied regarding data availability:

Agavekar, Gaurav; Hita Garcia, Francisco; P. Economo, Evan (2017): Taxonomic overview of the ant genus Tetramorium in India. figshare.

https://doi.org/10.6084/m9.figshare.c.3811468.v1

The following information was supplied regarding the registration of a newly described species:

Publication LSID: urn:lsid:zoobank.org:pub:5943B1C2-8978-4ECB-AB48-8ADB0A89E309.

Tetramorium jarawa sp. n. urn:lsid:zoobank.org:act:193548FC-C85D-4679-8B62-1F2BAF5B8699.

Tetramorium krishnani sp. n. urn:lsid:zoobank.org:act:1C00BC40-9AB1-4BA7-A58A-635951B13AC6.

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
