# Peer review of "Taxonomic overview of the hyperdiverse ant genus Tetramorium Mayr (Hymenoptera, Formicidae) in India with descriptions and X-ray microtomography of two new species from the Andaman Islands"

_PeerJ, doi:10.7717/peerj.3800_

## Round 0.1 · original submission · Minor Revisions

We ended up with 3 reviews here!, but all were positive about the work. Hopefully its easy to incorporate their suggestions into a revised ms. Great job on the paper.

Reviewer 1 ·

Basic reporting

No comment.

Experimental design

No comment.

Validity of the findings

I have the following small concerns about the descriptions. The authors described two new species, T. jarawa in T. inglebyi group and T. krishnani in T. tonganum group, and provided a key to species in each species group. Why did not the authors describe/redescribe the other species in the two species groups ? It is because the modified key implies the complete revision of the taxa. But, the authors presently recognize only 5 species in T. inglebyi group and 4 species in T. tonganum group. The key based on inadequate material might be sometimes confusing. Whereas, the two new species are very unique and easily separated from other species. I don't think that the authors need to provide a key in this paper.

Additional comments

The abstract needs more detail. I suggest that the authors should mention the number of Indian Tetramorium species. How many Tetramorium species are presently known from India ? If possible, the authors mention the species number for each species group (e.g., 42 species belonging to 12 species groups are recognized). They also state that "a key to the T. inglebyi group and T. tonganum group is provided/modified".

Introduction
line 94.
The authors should mention scientific significance of cybertypes although they referred to Hita Garcia et al (2017).

Identification key
line 327-328
The character states are unclear as criteria. The total number of ommatidia (facet) or the number in longest row should be used.

line 345-348
The key character for propodeal spines seems unclear. Can the authors show the relative length or index of propodeal spines ?

line 702
T. salvatum is in italic.

Figure 4
What does arrow in Fig. 4B indicate ?

Reviewer 2 ·

Basic reporting

In this study, the authors present a comprehensive work on the current taxonomic knowledge on the species rich ant genus Tetramorium in India. They present all relevant information about the 42 species of these ants currently known from India, an illustrated identification key to the species and a description of two species new to science.
Ants are used as study organism in a huge array of ecological studies, but many of these projects covering tropical and subtropical areas fail to identify a significant proportion of their studied ant to species level due to a lack of suitable identification literature. There this work is very imporantent to close this gap for this species rich ant genus in India.
The presentation of the study is excellent. The identification key is written and present very well, so that even non-experts on this ant genus can obtain proper results by using this key. This is achieved due to a thorought illustration of the relevant characters. The separation into several keys (a first key to the species groups and further keys to the single species groups) avoids a bulky key where user could get lost.
When it comes to the description of the species new to science, the authors provide high standards in the provision of a cybertype, a 3D image of the type species.
The readability of the text is very good, the authors used a clear English language which makes this manuscript easy to read.
My overall recommendation to the editors of PeerJ is to accept this manuscript for publication (with some minor revisions).

Experimental design

Concering the experimental design, I have some specific minor recommendations:

line 96 -99: location of studied material. It seems that all types (both holo- and paratypes) of the two new described species have been deposited at just a single location. In the light of risk management for taxonomic collections it might be advisable to deposite type material at several locations, preferable in different countries or world regions. Of course, I am aware that the deposition of such material is often ruled by political regulations and considerations of risk management by speading the material have a lower priority. But I would advise the authors to think about this point, maybe it should be possible to share he material at least between two collections based in India.

line 100-103: some basic information on the climate of the studied island?

Validity of the findings

no comments

Additional comments

Specific minor revisions

line 50 - 53: Maybe the authors could present the citations after the covered regions, e.g. "Afrotropical (citation of afrotropical study 1...), Malagasy(citation of study from Madagascar 1...)".

line 808: for a better comparison: please provide some numbers of other regions in Asia.

Figure 10. The quality of this figures is very low. The authors should try to improve this, maybe by taking own high resolution photographs of the specimens.

Figure 2B Map of Havelock Island: Two landcover types (agriculture/village and sand) almost look the same. This could be improved.

Figure 15. None of these pictures show a complete antenna of T. jarawa.

Data is deposited in Figshare: Pictures of Tetramorium jarawa: I would expect complete pictures. The dorsal view does not show the antennae and some parts of the head are lacking. This could be done better.

Reviewer 3 ·

Basic reporting

The paper is well-written and structured and provides an appropriate amount of literature references. I have included a few comments considering writing style in taxonomic descriptions in the general comments below.

Experimental design

The experimental design is appropriate for this study.

Validity of the findings

no comment

Additional comments

This study provides a review of the Indian Tetramorium fauna using material deposited in the National Center for Biological Sciences in Bangalore, India. Additionally, they did in depth sampling of a small island that is part of the Andaman archipelago. The authors describe two new species from the island and include an updated key to all species groups present in India. They also provide a species level key for the tonganum and inglebyi species groups which is where the newly described species are placed in. Moreover, the authors offer comments and notes on various misidentifications and confusions in the literature.

Overall, this is an important addition to our knowledge of Tetramorium in the region. The study is well-executed and I recommend publication after minor changes have been incorporated, see my comments below.

1. I suggest adding a little bit more detail into the abstract: 1) including species groups with the new species, 2) mentioning the species-level keys for the tonganum and inglebyi species groups
2. It would be helpful to the reader if the authors could include page references to 1) species group descriptions and 2) species level keys where present
3. Figures 4F and 4H as well as 4G and 4I would benefit from an arrow or another visual indication that shows the difference between these character states more clearly so the reader knows exactly what to look for – the differences are currently not very apparent.
4. I saw that all type specimens are currently deposited at the NCBS in India – I would suggest to send at least one type specimen to a second larger collection like for instance the London museum which has a large Tetramorium collection. I understand that for T. jarawa there are only 2 type specimens available – in this case, whether it makes sense to send one specimen off or not, must be evaluated by the authors.
5. I strongly suggest to avoid use of words like “comparatively” and “relatively” and “conspicuously” in taxonomic descriptions – unless you can provide an explanation that says what it is relative or comparative to (e.g., eye size in relation to scape width) OR if you can also provide absolute measurements (either mm or ommatidia number, in case of eyes). Examples from the text where this has been done well: 1) antennae of moderate length, not surpassing posterior head margin, 2) dorsum of petiolar node in dorsal view about as long as broad or clearly longer than broad. I understand that being specific is not always easy and the addition of images helps – however, I still strongly urge the authors to not solely rely on images but use descriptions as thoroughly as possible. Therefore, please review the text for any such words and examine if more specific descriptions can be provided. I included a few examples of the use of the above stated words in my line by line comments below.
6. Line by line comments follow:

45: Guenard reference seems formatted smaller than rest of text
176: punctuation missing
254: Mcglynn misspelled, should say McGlynn
325: conspicuously sculptured – absolute statement would be better
328: provide absolute measurement for eyes (comparing it to something else, giving mm or ommatidia number)
336: hairs comparatively short – provide absolute statement or compare to something specific
343: anterior clypeal margin entire – please consider if “clypeal margin complete” is more accurate [this appears on several occasions throughout the text]
345: comparatively
356: eyes moderate size – provide measurement [for all species group descriptions]
475: provide absolute measurement
478: absolute measurement
671: pilosity “simple” – provide absolute statement
687: not very clear sentence
739: typo “anterodorsam” and red comma present
800: word missing, should say “gastral tergite”
819: should say main distributions
829: should say “the Indian Tetramorium fauna”
837: What about those other undescribed species?
839: should say “the ant fauna”

---

## Round 0.2 · accepted · Accept

I am writing to inform you that your manuscript has been Accepted for publication.